## PHILOSOPHY OF BIOLOGY

# The analysis of living systems can generate both knowledge and illusions

**Abstract** Life relies on phenomena that range from changes in molecules that occur within nanoseconds to changes in populations that occur over millions of years. Researchers have developed a vast range of experimental techniques to analyze living systems, but a given technique usually only works over a limited range of length or time scales. Therefore, gaining a full understanding of a living system usually requires the integration of information obtained at multiple different scales by two or more techniques. This approach has undoubtedly led to a much better understanding of living systems but, equally, the staggering complexity of these systems, the sophistication and limitations of the techniques available in modern biology, and the need to use two or more techniques, can lead to persistent illusions of knowledge. Here, in an effort to make better use of the experimental techniques we have at our disposal, I propose a broad classification of techniques into six complementary approaches: perturbation, visualization, substitution, characterization, reconstitution, and simulation. Such a taxonomy might also help increase the reproducibility of inferences and improve peer review.

**ANTONY M JOSE\***

*For correspondence: amjose@umd.edu

**Competing interests:** The author declares that no competing interests exist.

## Introduction

Scientific knowledge relies on discoveries that continually extend, modify, or negate prior discoveries, making any inference from any study provisional. All practicing scientists intuitively appreciate this fact. Yet, collective blindness to alternative inferences could promote the acceptance of knowledge that later turns out to be unreliable. Such illusions are easily recognized in hindsight, but are difficult to anticipate. At an extreme, these illusions can provide a false sense of security about the foundations of a discipline, and also obscure the boundaries between the known and the unknown.

The pace of discovery and the array of techniques used in modern science make it difficult to evaluate new findings and place them in context. Modern biology is a good example: all biologists want to understand living systems, be they cells or populations or something in between, but the complexity of living systems and the sophistication of modern scientific enquiry can make it challenging to see the frontiers of the subject. Is it even possible to evaluate progress towards the overarching goals of any area of biology? A common framework for thinking about a wide range of experiments in biology could help us assess discoveries and their potential impact.

## Six approaches for analyzing living systems on multiple scales

One feature that many experimental techniques have in common, no matter how powerful they are, is that they are only useful over a limited range of length and time scales. Yet, the integration of information across different length and time scales is necessary to understand living systems because the outcome of events that occur at the level of molecules within nanoseconds and that unfold in populations over millions of years can depend on each other. For example, a mutation in DNA that disrupts molecular interactions could alter the cell, the tissue, the organism, and even populations. Conversely,

**Figure 1.** Six approaches for the analysis of living systems. In this schematic diagram, a living system is represented as an abstract network (center), with the colored nodes (circles) representing the different parts of the system, and the grey edges representing the interactions between the parts. Four of the six approaches described in this article involve doing something to one part of the system (shown here in black); the fifth approach involves combining multiple parts; and the sixth approach involves simulating some or all of the parts.

mechanisms that have been honed over evolutionary time in populations can repair the mutation in DNA or compensate for its consequence.

Here I argue that it is possible to classify the many different experimental techniques used in biology into six broad approaches (*Figure 1*): perturbation, visualization, substitution, characterization, reconstitution, and simulation. Each approach is applicable across all scales, ranging from molecules within an organism to populations of organisms within an ecosystem, which means that this framework can be used to evaluate any experiment and place it within the wider array of possible experiments. The six approaches are discussed in greater detail below, with an emphasis on analysis at the

molecular scale, along with some examples that illustrate application to other scales. Possible uses of this classification for improved inference and peer review are also discussed.

### 1. Perturbation

This approach involves changing something in a system and then observing it. What happens after a perturbation can depend not only on the part of the system being perturbed but also on how this part interacts with the rest of the system. Simplistic inference based on the presumed change in the part alone can be misleading. For example, loss of the *pha-1* gene in the worm *C. elegans* results in complete loss of the pharynx (throat) because all the cells of the pharynx fail to undergo morphogenesis and terminal differentiation (*Schnabel and Schnabel, 1990*). The initial simple model was that the PHA-1 protein is a 'master regulator' of pharynx differentiation. But subsequent analyses revealed that it is an antidote to a maternal toxin (*Ben-David et al., 2017*). Take away both the maternal toxin and PHA-1, and a pharynx is made just fine. Perturbation can be permanent or transient and can disrupt gene sequence (genetics), gene regulators (epigenetics), or the environment ('environomics'). In every case, the impact of the change depends on the changed part and the rest of the interacting system.

Prior conceptions about the ability of living systems to accommodate change can influence selection of perturbed systems for further analysis or indeed recognition that a perturbation has occurred. When performing grafting experiments on embryos to discover regions that cause changes when moved (*Mangold and Spemann, 1924*; see *Spemann and Mangold, 2001* for a translation), interpreting the results requires the idea that moving a piece of embryonic tissue could perturb patterning. Such perturbation is not guaranteed because of processes that can recover anatomy and function. For example, the entire organism can be generated from half the number of cells during regulative development (*Driesch, 1891*; see *Willier and Oppenheimer, 1964* for a translation), new cells can be made to regain pattern during regeneration (*Browne, 1909*), and cells can move to regain pattern during reorganization (*Abrams et al., 2015*). When using mutagens or chemicals to induce a particular change, unanticipated compensatory mechanisms can

confound inference (*El-Brolosy and Stainier, 2017*). A recent case that highlights this problem is the disagreement between morpholino-based perturbations and Cas-9-mediated genome editing in Zebrafish (*Kok et al., 2015*), some of which is currently thought to be explained by transcriptional adaptation in response to mutations (*Ma et al., 2019*; *El-Brolosy et al., 2019*). Compensation of any kind can blunt the impact of perturbations.

## 2. Visualization

Describing what can be seen in one or more parts of a system is often a first step towards understanding systems at any scale. Yet, limitations on what can be seen complicate inference. For example, describing an ecosystem requires accurate counts of organisms, which can be difficult to achieve and can distort deduced trophic relationships (reviewed in *Elphick, 2008*). Techniques for observing animal behavior without perturbation needed to be developed to reveal increases in the nocturnal activity of land animals (*Gaynor et al., 2018*) and light-based communication in deep-sea organisms (*Burford and Robison, 2020*). The advance of methods for seeing progressively smaller tumors illustrate how functionally important anatomical features of an organism could remain invisible until methods improve.

At the molecular scale, all visualization requires specialized instruments and techniques. The part to be seen can be either directly tagged with a label or indirectly through physical interactions with a label. Inference using these approaches requires the label to be similarly detectable in all parts of the system and the labeled parts to be used by the system in the same way as the unlabeled parts. A common tagging approach is the fusion of fluorescent protein or fluorescent RNA to a protein or RNA of interest (*Rodriguez et al., 2017*; *Truong and Ferré-D'Amaré, 2019*). However, because different cellular compartments can distort the signal from such fluorescent tags, specific optimization is needed for reliably detecting fluorescence in parts of a cell that differ in characteristics such as pH or oxidation state (*Costantini et al., 2015*). In addition, the fluorescent tag needs to be added to particular places on the part being visualized to avoid interfering with its functions. For example, the $G\alpha$ proteins of heterotrimeric G proteins interact with receptors, the beta-gamma subunits, regulators of G-protein signaling, and effectors, which leaves only a few places where a $G\alpha$ protein can be tagged without interfering with one of its functions (*Hughes et al., 2001*). Both how the system impacts the approach and how the approach impacts the system matter.

## 3. Substitution

This approach involves adding measurable proxies for parts of a system to infer how those parts are used by the system. For inference based on this approach to be successful, the proxy should not interfere with any aspect of the system while, at the same time, interacting with the system in the same way as the part, effectively substituting for the part in regulatory interactions. For example, one species can be used as a sentinel to report on the health of the ecosystem under the assumption that impacts on the sentinel species reflects impact on the ecosystem (*Popkin, 2020*). An inert tracer can be used as a proxy for other particles to understand how they move through a circulatory system (*Errico et al., 2015*). A reporter gene with the regulatory surroundings of a protein-coding gene can produce a visible proxy that can reveal how the system regulates the gene (*Chalfie et al., 1994*). In every case, if the proxy interferes with any aspect of the living system, a mix of perturbation and substitution has occurred, thus complicating inference.

Creating perfect proxies is not trivial and unexpected behaviors of proxies have revealed new aspects of how living systems function. For example, attempts to overexpress a gene for making a pigment in plants using many copies of the gene led instead to the silencing of the same gene expressed from the genome of the plant (*Napoli et al., 1990*; *van der Krol et al., 1990*) and the eventual discovery of RNA interference (*Fire et al., 1998*) – a potent method of silencing genes. Now we can assemble reporter genes with surgical precision using genome-editing approaches such as CRISPR (*Knott and Doudna, 2018*). Yet, permanent silencing of an apparently perfectly assembled proxy can occur through germline small RNAs in *C. elegans* for reasons that are not yet clear (reviewed in *Almeida et al., 2019*). Similarly, unexpected behaviors of proxies for parts involved in protein sorting, RNA splicing, or any other process of interest have the potential to reveal new aspects of the system rather than reliably reflecting the behavior of the parts. Sometimes a failed proxy teaches a new way to perturb the system.

## 4. Characterization

The purpose of this approach is to infer the properties of parts of the system. Organisms can be characterized as predator or prey based on their dietary habits to understand their influence on an ecosystem. However, some features of organisms can be less obvious and their eventual detection could warrant revisions of our understanding of the ecosystem (such as the recent detection of widespread biofluorescence in amphibians; *Lamb and Davis, 2020*). While the in vitro growth of organs (such as the optic cup of the eye; *Eiraku et al., 2011*) and cells (since the time of *Harrison, 1906*) can facilitate their detailed characterization, crucial aspects of their functions in vivo could be changed in vitro (for example, see *Bissell and Labarge, 2005*) for a discussion of the profound influence of in vivo microenvironments).

Similar considerations also plague methods for analyzing the properties of molecules. The shape of a protein can be inferred by solving its structure and the sequence of a genome can be inferred by interrogating its bases. These approaches typically require isolation of the parts or subsets of interacting parts and therefore do not reveal how the characterized parts interact with the rest of the system or whether aspects characterized in isolation are relevant for the system. For example, X-ray crystallography cannot be used to deduce the structures of regions of molecules that do not take on a rigid conformation (*Acharya and Lloyd, 2005*). Sometimes a characterization can turn out to be incomplete or wrong after a long period of time. For example, it took ~17 years to discover that a technique for determining if a cytosine on DNA is methylated (*Frommer et al., 1992*) does not discriminate 5-methyl cytosine from 5-hydroxymethyl cytosine (*Kriaucionis and Heintz, 2009*; *Tahiliani et al., 2009*). What we thought was one kind of modified base on DNA turned out to be an unknown mixture of two kinds of modified bases.

## 5. Reconstitution

This approach involves recovering particular properties of the system or subsystem by combining parts. Reconstitution can be performed at many different scales – from individual molecules being combined to recreate properties exhibited by collections of molecules, through cells being combined to recreate properties of organs or developing organisms, to organisms being combined to capture aspects of ecosystems. Recovery of a particular property of the system or subsystem by combining parts can provide compelling evidence for how the parts can function in the system. For example, the direct observation of F1-ATPase proteins rotating attached glowing actin filaments provided vivid support for a model for ATP production whereby the protein acts as a rotary motor (*Noji et al., 1997*). Interacting regulators can be combined to recreate behaviors in vitro, as in the case of the oscillations in phosphorylation recreated using three proteins and ATP (*Rust et al., 2011*). Similar assembly within cells can inform plausible regulatory logic of oscillatory gene expression (*Elowitz and Leibler, 2000*) – potentially a reconstitution of defined components (regulators) within poorly defined environments (cells).

Parts that are themselves living systems – such as cells – can be combined into tissues that develop into complex organs like the human brain when induced in vitro (*Lancaster et al., 2013*), highlighting the ability of cells to form highly structured communities. Likewise, the formation of human embryos from sperm and egg in vitro (*Edwards et al., 1969*) establishes the sufficiency of these two cell types. On a larger scale, multiple organisms can be grown together to recreate symbioses (such as *Vibrio fischeri* and its squid host *Euprymna scolopes*; *McFall-Ngai and Ruby, 1991*). However, in every case differences between the reconstituted system and the living system could persist either because of missing parts or erroneous arrangement of the parts. The progressive addition of such missing factors can recover increasingly sophisticated properties of the system (see *Ganzinger and Schwille, 2019*) for a review of this progression in understanding the cytoskeleton). Each missing property in a reconstitution experiment drives the discovery of additional factors to add towards the distant goal of reconstituting entire living systems of varying complexity from molecules.

## 6. Simulation

This approach is used to represent or abstract features of the system for analysis. Examples of simulations include: sketching ideas on a sheet of paper (such as Darwin's 1837 sketch of the branching diagram of descent with modification; *Barrett, 1959*); building models of molecular structures (such as the metal model of DNA; *Crick and Watson, 1954*); analyzing a real problem in abstract systems (such as the exploration of self-replication in cellular automata; reviewed

in *Sipper, 1998*); using computers to make quantitative or qualitative predictions of observed phenomena (such as computational models of cellular and molecular behaviors; reviewed in *Mogilner et al., 2006*). All simulations include some features and omit others.

Computer-aided simulations can explore the possible behaviors of systems under conditions that are experimentally unattainable using current technology to guide future experiments. Such calculations can span from sub-second molecular dynamics to multi-year experimental evolution. For example, the simultaneous behavior of multiple molecular complexes that regulate gene expression can be examined to test the validity of models aggregated from analyses on each complex. Such 'whole-cell' models can predict cell behavior despite being incomplete (*Karr et al., 2012*) and can be used to guide the experimental analysis of cells (*Sanghvi et al., 2013*). Further efforts in this direction hope to connect molecular detail to cell behavior, but require tremendous computational resources (see review by *Feig and Sugita, 2019*). At the other end of the scale, artificial 'organisms' that follow certain rules of evolution can be simulated in silico to gain intuition about the ways in which complexity could evolve (*Lenski et al., 2003*). However, recent surprises (*Phillips et al., 2019*) in the workings of the lac operon of *E. coli*, which has been the subject of more than half a century of experimentation and modeling (*Vilar et al., 2003*), underscore the need for iterative refinements of all in silico models for processes in living systems.

## Potential applications

This brief overview provides a glimpse into how the modern biologist seeks tentative conclusions using multiple powerful but imperfect approaches. Appreciation of the strengths and weaknesses of each approach could improve the reproducibility of inferences presented in a study, and could also increase fairness during peer review.

### Towards reproducible inference

Progress in science results from both exploratory and confirmatory analyses (*Tukey, 1980*), but exploratory analyses are often not reported (or are misleadingly reported as being confirmatory). Some advocate their disclosure as supplements to the main 'story' of the paper (*Thompson et al., 2020*) and others advocate obfuscating them if needed in service of improved story telling (*Sanes, 2019*). These opposing opinions about how science should be communicated are also relevant for establishing the desired reproducibility and rigor in science because problems can occur at the level of methods, results, or inferences (*Goodman et al., 2016*). Ways to improve the reproducibility of methods and results are being widely implemented, such as the inclusion of more detailed descriptions of experimental methods used and the statistical analyses performed. For inferences to be reproducible, qualitatively similar conclusions have to be drawn when a study is replicated or re-analysed (*Goodman et al., 2016*). However, it is not clear how the reproducibility of inferences can be improved, although the need for improvement is underscored by the results of a recent study which found that different teams of researchers drew different conclusions from the same neuroimaging dataset (*Botvinik-Nezer et al., 2020*).

Published inferences can be preceded by continuous interplay of exploratory and confirmatory research. As a result, the number of possible inferences explored in a study is typically not carefully documented. Furthermore, there can always be unimagined additional inferences of varying probabilities. This difficulty in deciding 'N' for plausible inferences makes it impossible for a practicing scientist to know what 'statistical test' they could use to communicate the reasonableness of their collective inferences. Furthermore, language could exacerbate this difficulty because persuasive story telling can make it difficult to spot gaps in logic. Although critical linguistic analysis has been proposed as a way to ameliorate this problem (see *Segal, 1993*; *Horton, 1995*) and the ensuing discussions), it has not been widely implemented. In short, the expected reproducibility of inferences is difficult to quantify because of the way science is done and is obscured because of the way science is communicated.

Inferences that are drawn from a study are influenced by prevailing paradigms. Erroneous paradigms can drive the accumulation of wrong inferences for long periods, as famously occurred with the Ptolemaic model of our solar system. Such episodes illustrate how knowledge derived from perfectly reproducible observations can nevertheless be illusory. The scientific community could collectively decide that such irreproducible inferences are part of progress – each one a potential impetus for a paradigm shift (*Kuhn, 1962*) to be eventually realized by some discerning scientist(s). Alternatively, we

could explore ways to reduce the time and resources spent taking such missteps. In either case, having an explicit overview of the basis for the overall inferences and conclusions drawn in a study could clarify the next experiments and stimulate exploration of alternative inferences.

Currently, inferences are discussed in the introduction and discussion sections of papers: however, it would be much better if papers included explicit statements about the rationales for the inferences drawn by the authors (as is currently done for experimental methods and statistical analyses). This could be done in a relatively straightforward manner by adding two elements to the methods section of a paper: i) an explicit statement of the prevailing paradigm or prior model as understood by the authors; ii) a summary of the basis for any proposed modification of the model in plain language (for example, 'Inferences for the proposed model were drawn based on a perturbation approach using two techniques with distinct caveats and a visualization approach using one technique'). Finally, a common taxonomy of approaches with acknowledged limitations, refined through periodic debates, could help outsiders develop their own critical overviews of progress in a field and help the next generation of scientists decide on their future research directions.

### Towards fair peer review

Authors and reviewers debate about methods, results, and inferences during peer review. In such debates, the approaches outlined in this article could provide a common context for articulating strengths and weaknesses. Such structured evaluation could benefit both peer review of a particular study and review articles that appraise a field to identify areas of future development. Since all doubt about the conclusions of any study cannot be eliminated, a commonly agreed upon level of support for claims is needed. For example, it would be unreasonable to expect any study to include experiments from all six approaches. Moreover, since a study that uses a single experimental approach well could be better than a study that uses two or more approaches poorly, statements about quantity and quality need to be developed. Such explicit statements about the acceptable level of doubt for publication in a particular journal, or for receiving a grant from a particular funding agency, could guide reviewers and authors through the process of peer review.

Peer review by journals or funding agencies also involves estimating the impact of the study.

Since the ultimate value of any study is established through follow up work by the community at large, any estimate of potential impact made during peer review is at best an educated guess influenced by prevailing paradigms (*Park et al., 2014*; *Siler et al., 2015*). Furthermore, despite agreeing upon a common level of support for claims, there can be disagreements because of subjective interpretations (for example, see *Keeling et al., 2019* for a discussion of the different meanings of the word 'function' in biology). This subjectivity is underscored by studies on grant review that document low agreement when different panels judge the same proposal (*Pier et al., 2018*). Given all this, being more specific about the paradigms involved in a paper or grant application – both in the paper or application itself, and during the review and assessment of the paper or application – should lead to better decisions being made.

Experiments are underway on the process of peer review (*Rennie, 2016*; *Rodgers, 2017*; *Polka et al., 2018*), with a number of journals opting to publish all exchanges between authors and reviewers to improve transparency (see, for example, *Nature, 2020*). These exchanges are typically between experts and can be more abstruse than the paper being reviewed, making it difficult for the interested reader to decipher. One solution is to provide a summary of the discussion as is already done by some funding agencies. The editor of a manuscript could guide readers by summarizing the major points of debate during its review: ideally, this summary would be written in plain language (e.g., 'The need for a visualization approach, caveats of two experimental techniques, and plausibility of alternative inferences were debated'). eLife recently took a step in this direction with the introduction of 'acceptance explanations'. The exchanges that take place between authors and reviewers during peer review provide a genuine opportunity for readers to learn about the process of science.

## Conclusion

Biologists are engaged in understanding living systems through experimental and theoretical efforts in many subdisciplines. Placing new advances in the context of overall progress in biology can be challenging because of the dazzling variety and sophistication of techniques used. The six complementary approaches presented here provide a scaffold for classifying techniques and contextualizing discoveries.

Appreciating the scope and limitations of each approach could improve how inferences are drawn and how studies are reviewed. Together, better inference and better review could promote the realistic appraisal of discoveries and limit persistent illusions of knowledge.

## Acknowledgements

I thank Tom Kocher, Karen Carleton, Charles Delwiche, and members of the Jose lab for long discussions; and Tom Kocher, Karen Carleton, members of the Jose lab, the editor and the reviewers for their comments on the manuscript.

**Antony M Jose** is in the Department of Cell Biology and Molecular Genetics, University of Maryland, College Park, United States
amjose@umd.edu
https://orcid.org/0000-0003-1405-0618

*Author contributions:* Antony M Jose, Conceptualization, Writing - original draft, Writing - review and editing

*Competing interests:* The author declares that no competing interests exist.

### Funding

| Funder | Grant reference number | Author |
| --- | --- | --- |
| National Institutes of Health | R01GM111457 | Antony M Jose |
| National Institutes of Health | R01GM124356 | Antony M Jose |

The funders had no role in study design, data collection and interpretation, or the decision to submit the work for publication.

**Decision letter and Author response**
Decision letter https://doi.org/10.7554/eLife.56354.sa1
Author response https://doi.org/10.7554/eLife.56354.sa2

## Additional files

### Data availability

No data were generated for this work.

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
