## [Decision Letter]

Thank you for submitting your article "Knowledge and illusions when using the complementary approaches needed to understand living systems" to *eLife* for consideration as a Features Article. Your article has been reviewed by three peer reviewers, and I am writing to invite you to revise the article in response to the comments of these reviewers.

The points that need to be addressed in the report of reviewer #1 are clear to see. In the report from reviewer #2, please address the points in the second and fourth paragraphs; please also take on board the comments in the third paragraph about making the article more readable (but also note that if the article is accepted for publication, it will be edited for readability and clarity). As regards the report from reviewer #3, it may be difficult to answer the questions in the second paragraph, but addressing the points raised by reviewer #1 will help to address some of these concerns.

Reviewer #1

This Essay addresses methodological and philosophical aspects of Biology broadly conceived. The author asks important questions, including "How is an outsider to evaluate [biologists'] progress?" (although I'm not sure a solid answer is proposed here), and gives a classification (systematization) of the major approaches in the field. This and a number of other important meta-scientific questions, including reproducibility and fairness of review, are discussed. Overall this is an important approach and I think will be valuable, but it seems a little thin on actionable suggestions (except for the good and novel idea to be explicit in papers about inference and method with respect to a common taxonomy), or on deep insights that hadn't been made in longer pieces in the past (admittedly, in journals that few biologists read). I suggest some points which will increase value for readers.

The title might use a little work - I'm not sure everyone will understand the meaning of "illusions" (without having read the paper first), but it may draw people in so perhaps it's fine. Overall I find the arguments convincing. A few minor comments:

- "Inference using [perturbation] approaches requires prior models of how the living system is built." I am not sure this is right - many perturbation experiments (e.g., classic work in developmental biology using transplantation, teratology, etc.) were done with absolutely no model of mechanism, and even modern screening (high-throughput) approaches are often done in a model-agnostic manner. Overall I suggest that the Perturbation section not be exclusively focused on genetics, as this is just one of the perturbational tools in the biologists' toolbox. It's too narrowly conceived as written, to match the title of the piece which refers to "living systems" (i.e., not just the molecular level).

- "Visualization" is also focused entirely on molecular tagging. I think if the author wants to stick to this level of analysis, the title might need to be made more specific - right now, it promises a more comprehensive view of biology. Lots of aspects of visualizing systems on the anatomical, metabolic, neurobiological, behavioral, and population levels are not even mentioned here.

- "Reconstitution" is likewise envisioned purely molecularly, not including for example important reconstitution tools like grafts/chimeras, organoids, biobots, etc. and other examples of synthetic bioengineering.

- Applications. The author is right to focus some spotlight on exploratory analyses, which are often hidden in favor of 'hard inference' storytelling, but not much is said here about what to do about this.

Overall, the task of trying to systematize biology in these ways is valuable and I applaud the author for taking this on. However, as written, it's very narrowly construed and I'm not sure it provides the insights for all of Biology that the introductory text suggests, or that *eLife* readers will want. One way to resolve this is to put "Molecular Biology" or similar in the title, and simply decide that this will be only about that level of investigation. On the other hand, if there is room, I would be happier to see a bigger piece that lived up to the much more fundamental and inclusive aim suggested in the Abstract: "acquire knowledge about living systems". As it is now, it's far too narrow for that (as large a goal as that is).

Reviewer #2

I have found this a very difficult paper to review. Normally as a reviewer I try to assess the extent to which the methodology and results are rigorous and whether the underpinning literature is treated with appropriate scholarship, then whether the conclusions are justified (or at least the speculations reasonable) based on the information presented. But this manuscript is not a scientific paper but an essay on epistemology, therefore an "opinion" piece. Therefore I cannot use my usual method to assess it. I can only answer with more "opinion" and I generally shy away from doing this as this is the realm of religion and politics, not Science.

Overall I do feel that this piece does raise some interesting points that are worthy of further discussion and consideration by many scientists. I am not sure I agree that this is a definitive classification of approaches (for example simple "description" of the system at a chosen level or levels is not obviously included and many studies start this way, and there are various types of theoretical model building that are not easily included under "simulation") but the six chosen ones for functional analysis are common ones.

Whether this is the type of paper that *eLife* wants to publish, and under what heading, is a matter for the editors. I think it belongs more as an editorial in a journal like Nature or Science, or perhaps as an Essay in BioEssays or similar. In either case it needs to be written much more fluidly to make it more attractive and accessible to experimental scientists who are not otherwise inclined to think about these bigger philosophical and epistemological issues. I find this quite difficult to read even as someone who has been thinking about these issues for some time.

I don't like the proscriptive tone in relation to what reviewers should do - it gives an unnecessarily arrogant tone to the essay. While it is true that being aware of the strengths and limitations of particular types of approaches is essential to determine whether the conclusions of a set of experiments are justified, there are many different ways for reviewers to achieve this and this six-part classification is too rigid a framework to advocate for general use. I think the paper would gain if the comments about peer review were removed. Where this classification (or other ways of evaluating evidence and conclusions) is probably particularly important is in the writing of reviews that synthesize the work of a number of papers in a field - many reviews are not sufficiently analytical or integrative and therefore often fail to undertake a critical evaluation that could allow readers to see the strengths and limitations of different studies more clearly.

Reviewer #3

This is an unusual paper for me to review and I offer my reactions to it – I am not sure whether what I have to say is in any objective way right or wrong. The author makes some interesting observations about how scientists use perturbation, visualization, substitution, characterization, reconstitution, and simulation of some aspect of a living system in an effort try to understand how it works. The author emphasizes how each of these approaches in isolation, coupled with sociological norms of scientists have contributed to erroneous or misleading conclusions, usually using one spectacular example to make the point for each approach in this essay.

Its not clear to what degree these examples are representative of the broader life science enterprise, though its clear they have played a role in sustaining some misconceptions for a long time. Is there some way in which the significance and frequency of examples used can be put into perspective?

The issues the author raises are certainly worth considering, and an essay that brings them to the reader's attention is important. However, it is not clear how effective the suggested solution of explicitly spelling out the assumed paradigm will be. Other suggestions for the editor in capturing the discussions and making this a more transparent process are good – and already standard practice for *eLife*!

Communicating more of the backstory behind a study including the blind alleys, alternate hypotheses and sometimes accidental discovery of the appropriate framework would also be helpful and interesting. Most scientists are quite aware of the contrived nature of the narrative that is often created to make a concise slick paper with few unresolved open questions. Have space limitations and the perceived desires of some "premier" journals (at least in the past) played some role in encouraging the evolution of this contrived narrative? Perhaps digital publishing provides the opportunity to more easily respond to the issues emphasized in this essay, with opportunities to provide transparency and multiple opportunities to dig deeper through supplementary material into the back story, alternative hypotheses, statistical analysis etc.

---

## [Author Response]

[We repeat the reviewers’ points here in italic, and include our replies point by point, as well as a description of the changes made, in Roman.]

Reviewer #1This Essay addresses methodological and philosophical aspects of Biology broadly conceived. The author asks important questions, including "How is an outsider to evaluate [biologists'] progress?" (although I'm not sure a solid answer is proposed here), and gives a classification (systematization) of the major approaches in the field. This and a number of other important meta-scientific questions, including reproducibility and fairness of review, are discussed. Overall this is an important approach and I think will be valuable, but it seems a little thin on actionable suggestions (except for the good and novel idea to be explicit in papers about inference and method with respect to a common taxonomy), or on deep insights that hadn't been made in longer pieces in the past (admittedly, in journals that few biologists read). I suggest some points which will increase value for readers.

I thank the reviewer for the positive words and for endorsing explicit inclusion of statements about inferences in the Methods section. Where possible, I have made suggestions for how one might address problems. However, it is possible that some problems are not easily solvable at the present time potentially because of underdeveloped methods (e.g., evaluating the impact of conflated exploratory and confirmatory research). In such cases, I have merely drawn attention to the problem and highlighted the current difficulty.

The title might use a little work - I'm not sure everyone will understand the meaning of "illusions" (without having read the paper first), but it may draw people in so perhaps it's fine. Overall I find the arguments convincing.

I have simplified the title to be “Knowledge and illusions acquired during the analysis of living systems”.

Editor's note: The title was later changed to: "The analysis of living systems can generate both knowledge and illusions"

A few minor comments:

Collectively, the comments below suggest an expansion of the scope of the article to include the analysis of living systems on multiple scales. Although the taxonomy can apply across multiple scales (reason for representing living systems as an abstract network in Figure 1), the previous version did not highlight this aspect and was focused on a single scale - molecular - for simplicity. I have now highlighted the multi-scale nature of the proposed taxonomy during the early part of the article and provided specific examples to address other scales throughout the article as suggested by the reviewer.

- "Inference using [perturbation] approaches requires prior models of how the living system is built." I am not sure this is right - many perturbation experiments (e.g., classic work in developmental biology using transplantation, teratology, etc.) were done with absolutely no model of mechanism, and even modern screening (high-throughput) approaches are often done in a model-agnostic manner. Overall I suggest that the Perturbation section not be exclusively focused on genetics, as this is just one of the perturbational tools in the biologists' toolbox. It's too narrowly conceived as written, to match the title of the piece which refers to "living systems" (i.e., not just the molecular level).

I thank the reviewer for pointing out this error. This statement was meant to draw attention to the prior conception of the living system that is necessary to infer what a perturbed outcome means or indeed that a perturbation has occurred. I agree that ‘model’ is not a good word for conveying this idea because it connotes molecular mechanism in the context of the rest of the article (previous version). I have now expanded upon the statement to make the intended meaning clearer. I have also included some examples of the different kinds of perturbation suggested by the reviewer and illustrated the particular prior conceptions that impact the inference after a perturbation experiment. For example, in the famous Spemann and Mangold experiments on the organizer in embryology, interpreting the results requires the idea that a piece of embryonic tissue *could* be the source of unknown factors that help pattern the entire embryo. Modern screening selects for particular perturbed outcomes based on the prior expectation of what a perturbed system of interest would look like or that a particular manipulation is a perturbation, which sometimes lead to unexpected outcomes. For example, genetic compensation can confound inference (El-Brolosy and Stainier, *PLoS Genetics*, 2017). A prominent recent case that highlights this problem is the disagreement between morpholino-based perturbations and Cas-9-mediated genome editing in Zebrafish (Kok et al., *Dev. Cell*, 2015), some of which is currently thought to be explained by transcriptional adaptation (Ma et al., *Nature*, 2019, El-Brolosy et al., *Nature*, 2019).

- "Visualization" is also focused entirely on molecular tagging. I think if the author wants to stick to this level of analysis, the title might need to be made more specific - right now, it promises a more comprehensive view of biology. Lots of aspects of visualizing systems on the anatomical, metabolic, neurobiological, behavioral, and population levels are not even mentioned here.

I have now used examples from multiple scales to illustrate the kinds of visualizations that are possible and the kinds of limitation that can confound inference. For example, at the anatomical scale, limitations in the resolution of measurement combined with preconceptions promoted by prevailing theories can result in missed features of the system – e.g., consider estimations of tumor size and the developments that have been necessary to improve their visualization. At the population scale, difficulties in counting a particular species can distort the deduced composition and trophic relationships in an ecosystem. At the behavioral level, inability to visualize aspects of behavior can result in profound misconceptions about an organism. For example, recent ability to see animal behavior at night in the wild without perturbation is changing our understanding of land animals. In the deep ocean, recent ability to image without perturbation has revealed light-based communication in benthic organisms.

- "Reconstitution" is likewise envisioned purely molecularly, not including for example important reconstitution tools like grafts/chimeras, organoids, biobots, etc. and other examples of synthetic bioengineering.

I have now expanded this section to include reconstitution on additional scales. For example, human embryos can be formed from sperm and egg in vitro (Edwards, et al., 1969), establishing the sufficiency of these two cell types for the formation of human embryos.

- Applications. The author is right to focus some spotlight on exploratory analyses, which are often hidden in favor of 'hard inference' storytelling, but not much is said here about what to do about this.

The complications introduced by the conflation of exploratory and confirmatory research when story telling is emphasized require a deeper analysis and debate by the scientific community that are beyond the scope of this article. Unfortunately, a facile solution for this problem may not be available. Nevertheless, widespread awareness of the problem is the starting point for collectively working towards a solution. Therefore, the primary goals of this article remain to introduce a unified classification of approaches to understand living systems and invite the scientific community to debate the ways to implement such system-level and meta-scientific thinking to improve the practice of science.

Overall, the task of trying to systematize biology in these ways is valuable and I applaud the author for taking this on. However, as written, it's very narrowly construed and I'm not sure it provides the insights for all of Biology that the introductory text suggests, or that eLife readers will want. One way to resolve this is to put "Molecular Biology" or similar in the title, and simply decide that this will be only about that level of investigation. On the other hand, if there is room, I would be happier to see a bigger piece that lived up to the much more fundamental and inclusive aim suggested in the Abstract: "acquire knowledge about living systems". As it is now, it's far too narrow for that (as large a goal as that is).

I have now expanded the scope of the article to highlight the application of the taxonomy to additional scales and included illustrative examples across scales throughout the article.

Reviewer #2I have found this a very difficult paper to review. Normally as a reviewer I try to assess the extent to which the methodology and results are rigorous and whether the underpinning literature is treated with appropriate scholarship, then whether the conclusions are justified (or at least the speculations reasonable) based on the information presented. But this manuscript is not a scientific paper but an essay on epistemology, therefore an "opinion" piece. Therefore I cannot use my usual method to assess it. I can only answer with more "opinion" and I generally shy away from doing this as this is the realm of religion and politics, not Science.

As noted by reviewer #1, this article addresses methodological and philosophical aspects of science, particularly as it applies to understanding living systems. The spirit of the article is in keeping with the belief that philosophical considerations can have practical uses for science (e.g., Laplane et al., PNAS, 2019). Methodologically, this article develops an overarching classification and invites future debates about such overall classifications. Philosophically, this article highlights the ever present unknown unknown when using any approach that can cast doubt over the most ardently held knowns.

Overall I do feel that this piece does raise some interesting points that are worthy of further discussion and consideration by many scientists. I am not sure I agree that this is a definitive classification of approaches (for example simple "description" of the system at a chosen level or levels is not obviously included and many studies start this way, and there are various types of theoretical model building that are not easily included under "simulation") but the six chosen ones for functional analysis are common ones.

As stated in the article, future periodic debates about the classification would help enrich and sharpen our understanding of the methods we have and will acquire. The previous version of the article did not expand upon application of the taxonomy across multiple scales, which makes it unclear where “description” fits within the classification. As suggested by reviewer #1, I have now expanded upon the multi-scale application, and “description” at any scale is essentially “visualization”, i.e., seeing what is there and reporting on it. For example, when describing an ecosystem, we count the number of organisms, look at their shape, size and any other characteristics that we can see (using the naked eye or using additional instruments). In the current expanded section on “simulation”, methods ranging from the sketching of ideas on a sheet of paper to simulate how a system might be working to running in silico computations to explore a virtual recreation of a system are included.

Whether this is the type of paper that eLife wants to publish, and under what heading, is a matter for the editors. I think it belongs more as an editorial in a journal like Nature or Science, or perhaps as an Essay in BioEssays or similar. In either case it needs to be written much more fluidly to make it more attractive and accessible to experimental scientists who are not otherwise inclined to think about these bigger philosophical and epistemological issues. I find this quite difficult to read even as someone who has been thinking about these issues for some time.

I have strived to simplify sentences throughout the manuscript and appreciate the editorial help offered by the journal for improving clarity and readability. I hope that these efforts make the article more engaging.

I don't like the proscriptive tone in relation to what reviewers should do - it gives an unnecessarily arrogant tone to the essay. While it is true that being aware of the strengths and limitations of particular types of approaches is essential to determine whether the conclusions of a set of experiments are justified, there are many different ways for reviewers to achieve this and this six-part classification is too rigid a framework to advocate for general use. I think the paper would gain if the comments about peer review were removed. Where this classification (or other ways of evaluating evidence and conclusions) is probably particularly important is in the writing of reviews that synthesize the work of a number of papers in a field - many reviews are not sufficiently analytical or integrative and therefore often fail to undertake a critical evaluation that could allow readers to see the strengths and limitations of different studies more clearly.

The intent of the suggestions is not to be proscriptive, but rather to add a specific suggestion that can either be taken on or ignored after debating its merits. I have now reworded the section to mitigate any perceived arrogance in tone. As highlighted in the article, the process of peer review is currently undergoing a lot of experimentation and this suggestion adds to the debate on the best way to achieve peer review, which in my opinion is an invaluable part of science. I believe that constructive peer review can improve an author’s work as it certainly has for this article.

I agree with the excellent opinion of the reviewer that this classification would help with the writing of more analytical and integrative reviews, and the revised manuscript now includes a sentence to highlight this use in the peer review section.

Reviewer #3This is an unusual paper for me to review and I offer my reactions to it-I not sure whether what I have to say is in any objective way right or wrong. The author makes some interesting observations about how scientists use perturbation, visualization, substitution, characterization, reconstitution, and simulation of some aspect of a living system in an effort try to understand how it works. The author emphasizes how each of these approaches in isolation, coupled with sociological norms of scientists have contributed to erroneous or misleading conclusions, usually using one spectacular example to make the point for each approach in this essay.

I thank the reviewer for the positive words.

Its not clear to what degree these examples are representative of the broader life science enterprise, though its clear they have played a role in sustaining some misconceptions for a long time. Is there some way in which the significance and frequency of examples used can be put into perspective?

The reviewer raises an excellent and difficult question, answering which requires extensive study that is beyond the scope of this article. Specifically, the answer needs a comprehensive historical (and predictive?) survey of the science that has been done (and will be done?). One source that presents a partial historical survey of examples where misconceptions arose, persisted for a long time and then were overthrown is the book ‘Structure of Scientific Revolutions’ by Thomas Kuhn. However, even such a study doesn’t allow evaluation of the frequency. Evaluation of the frequency is being made all the more difficult by the growing volume of scientific literature.

In the revised version of the article, application of the taxonomy to additional scales and subdisciplines are highlighted in an effort to underscore the broad use of such a taxonomy of methods across the life sciences (and perhaps all sciences).

The issues the author raises are certainly worth considering, and an essay that brings them to the reader's attention is important. However, it is not clear how effective the suggested solution of explicitly spelling out the assumed paradigm will be. Other suggestions for the editor in capturing the discussions and making this a more transparent process are good -and already standard practice for eLife!

Explicitly spelling out the assumed paradigm can help authors and readers identify the assumptions/frameworks that are necessary for the inferences to be valid. Often these remain unstated and could result in quests for concepts and entities that may not exist – historical examples include the search for luminiferous ether as the medium transmitting light and phlogiston as the carrier of heat. Nevertheless, the suggestions presented in the article are starting points for future debate. The primary goal of the article is to bring attention to these issues that impact how science is done, presented, and evaluated.

Unlike *eLife*, many journals do not yet have active summaries of the review process by the editors. I hope the arguments presented in this article add to those presented elsewhere to aid wider adoption of this practice.

Communicating more of the backstory behind a study including the blind alleys, alternate hypotheses and sometimes accidental discovery of the appropriate framework would also be helpful and interesting. Most scientists are quite aware of the contrived nature of the narrative that is often created to make a concise slick paper with few unresolved open questions. Have space limitations and the perceived desires of some "premier" journals (at least in the past) played some role in encouraging the evolution of this contrived narrative? Perhaps digital publishing provides the opportunity to more easily respond to the issues emphasized in this essay, with opportunities to provide transparency and multiple opportunities to dig deeper through supplementary material into the back story, alternative hypotheses, statistical analysis etc.

I agree with the reviewer that digital publishing can facilitate the clear declaration of the often tortuous route to discovery. However, our collective inability to evaluate the statistical rigor of the inferences in the study would remain. Furthermore, pursuit of today’s ‘complete’ story while ignoring or obfuscating incompleteness, deliberately makes scientific understanding more saltatory than it needs to be – proceeding from one false picture to the next and reducing confidence in published models. The article therefore highlights this difficulty to invite debate on the most productive courses of action for the scientific community.